# Detection of Microplastic in Human Placenta and Meconium in a Clinical Setting

**DOI:** 10.3390/pharmaceutics13070921

**Published:** 2021-06-22

**Authors:** Thorsten Braun, Loreen Ehrlich, Wolfgang Henrich, Sebastian Koeppel, Ievgeniia Lomako, Philipp Schwabl, Bettina Liebmann

**Affiliations:** 1Clinic of Obstetrics, Charité—Universitätsmedizin Berlin, Corporate Member of Freie Universität und Humboldt-Universität zu Berlin, Augustenburgerplatz 1, 13507 Berlin, Germany; wolfgang.henrich@charite.de; 2Division of Experimental Obstetrics, Charité—Universitätsmedizin Berlin, Corporate Member of Freie Universität und Humboldt-Universität zu Berlin, Augustenburgerplatz 1, 13507 Berlin, Germany; loreen.ehrlich@charite.de; 3Environment Agency Austria (Umweltbundesamt GmbH), 1090 Vienna, Austria; Sebastian.Koeppel@umweltbundesamt.at (S.K.); Ievgeniia.Lomako@umweltbundesamt.at (I.L.); bettina.liebmann@umweltbundesamt.at (B.L.); 4Division of Gastroenterology and Hepatology, Department of Internal Medicine III, Medical University of Vienna, 1090 Vienna, Austria; philipp.schwabl@meduniwien.ac.at

**Keywords:** microplastics, placenta, fetus, pregnancy, polystyrene

## Abstract

Environmental pollution with microplastics (MPs) is a major and worldwide concern. Involuntary exposure to MPs by ingestion or inhalation is unavoidable. The effects on human health are still under debate, while in animals, cellular MP translocation and subsequent deleterious effects were shown. First reports indicate a potential intrauterine exposure with MPs, yet readouts are prone to contamination. Method: To establish a thorough protocol for the detection of MPs in human placenta and fetal meconium in a real-life clinical setting, a pilot study was set up to screen for MPs > 50 µm in placental tissue and meconium sampled during two cesarean sections for breech deliveries. After chemical digestion of non-plastic material, Fourier-transform infrared (FTIR) microspectroscopy was used to analyze the presence of 10 common types of microplastic in placenta and stool samples. Results: Human placenta and meconium samples were screened positive for polyethylene, polypropylene, polystyrene, and polyurethane, of which only the latter one was also detected as airborne fallout in the operating room—thus representing potential contamination. Conclusion: We found MPs > 50 µm in placenta and meconium acquired from cesarean delivery. Critical evaluation of potential contamination sources is pivotal and may guide future clinical studies to improve the correct detection of MPs in organ tissue. Studies investigating nano-sized plastics in human tissue are warranted.

## 1. Introduction

Environmental pollution with microplastic particles (MPs) is omnipresent. MPs were found not only in urban surroundings but also deep in the oceans, up in the mountains, and even in the arctic snow [1,2,3,4]. According to a recent consensus statement, MPs are defined as “synthetic solid particles or polymeric matrices, with regular or irregular shape and with size ranging from 1 µm to 5 mm, of either primary or secondary manufacturing origin, which are insoluble in water.” [5].

MPs have been shown to enter the food chain [6] and consequentially were also detected in human stool [7]. The estimated yearly MP consumption ranges from 39,000 to 52,000 particles [8]. Thus far, MPs have been mainly detected in water, teabags, baby bottles, shellfish, and salt [6,8,9,10,11]. Hence, it appears that involuntary ingestion is unavoidable. Another potential entry point of MPs into the human organism is via inhalation, as MPs are also commonly present as airborne particles [12]. While the effects of MPs on human health are still under debate [4,13], animal studies prove the translocation of MPs into organs, which subsequently causes deleterious effects [14,15,16]. Physical properties (size, shape, and length), chemical properties (presence of additives and polymer type), particle concentrations, and microbial films are key determinants affecting MP toxicity [4].

In humans, the uptake of micro- and especially nano-plastics seems plausible since particles smaller than 150 µm can cross the gastrointestinal epithelium in mammals [4]. However, scientists speculate that only 0.3% of these particles are expected to be absorbed [6]. Especially, particles with a size of <10 µm are likely to cross cellular membranes and thus may also enter the placenta [4], ultimately posing a potential risk for the unborn fetus [17].

In earlier studies, Grafmueller et al. demonstrated the ability of polystyrene nanoparticles to cross the placental barrier [18,19]. This placenta permeability was further confirmed with carbon particles ranging from 1 to 10 µm [20]. Recently, transplacental movement of nanoplastic particles (<100 nm) has been demonstrated using ex vivo placenta perfusion models [6,21,22], which was accompanied by a change in the protein corona composition [23]. Moreover, the cellular uptake and intracellular accumulation of nano- and microparticles have been shown in placenta in vitro co-culture models [24]. Ultimately, in a very recent paper, the presence of MPs ranging from 5 to 10 μm was detected in out of 5 placentas, delivered vaginally [25].

Since contamination of stool (which has been proven to contain MPs [7]) during vaginal birth is common and as the clinical setting of a delivery room needs to be considered as potential sources for contaminants, we aimed to investigate the presence of MPs > 50 µm in human placenta derived from cesarean delivery including a thorough analysis of potential contamination sources in order to refine the methods of MP detection in a clinical setting (Figure 1). Additionally, in order to study potential MPs ingestion of the fetus (suggesting transplacental MP movement), we also quantified MP concentrations in meconium.

## 2. Materials and Methods

### 2.1. Study Population

The present pilot study on term placenta and meconium enrolled two mothers giving birth via cesarean delivery with singleton pregnancies and breech presentation. Both mothers gave birth at the Charité University Berlin, Germany, and gave informed consent to participate in this pilot trial as part of another study, which was approved by the Ethics Committee of the Charité University (EA4_059_16, 8 December 2016). The study was conducted according to the guidelines laid down in the Declaration of Helsinki. Both women gave their written and informed consent to the study. Mothers were asked to fill out a questionnaire as previously used [7] to assess lifestyle information.

### 2.2. Sample Collecting 

The cesarean delivery was performed according to current medical guidelines [26]. The initial sampling (phase 1) comprised of (ID: A) peripheral and (ID: B) central placenta tissue, meconium (ID: D), and maternal stool (ID: F). Respective negative controls (IDs: C, E, G) comprised of the preparation instruments and storage containers without actual human tissue samples (Figure 2). A wide range of potential contamination controls (IDs: 1–16) was collected in the operating theatre. After uterotomy and delivery of the rump of the fetus, meconium (ID: D) spontaneously emptied from the bowel and was collected under sterile conditions and transferred into prepared cleaned glass bottles within the operating theatre. Instruments used for this purpose were stored separately in a metal tray. The placenta (IDs: A, B) was delivered spontaneously without manual removal and transferred into a prepared sterile metal kidney shell. In order to not touch the placenta, it was born by gentle cord traction and held above the metal kidney shell before cutting the cord with a metal scissor and dropping the placenta into the shell. Maternal stool samples (ID: F) were collected after birth according to previously published protocols without plastic materials and transferred into cleaned glass bottles using a metal spatula [7]. Negative controls were sampled in the lab (IDs: C, E, G). The contamination controls included materials used during the cesarean delivery: surgical mask, head cover, drape, pad, lab sponge green, white swab, soaking drape, packaging of drape, gauze ball, packaging of gauze ball, table protection, sterile glove, gown, scrubs, tubing, and non-sterile gloves (IDs: 1–16).

After reviewing results from the initial investigations from patient one, we undertook a refined sampling procedure (phase 2) in the second patient. From the placenta, one 1 × 1 × 1 cm piece was taken after intense washing (Millipore SAS, 67120 Molsheim, France) (ID: H), and another 1 × 1 × 1 cm piece was excised from the core tissue (ID: I). Again, we sampled meconium (ID: K). Additionally, in this phase, airborne fallout (ID: M) was investigated as potential contamination. For this purpose, microplastic-free collection containers with an open lid were positioned in the operating room for 10 min and then closed again. The analytical process in phase 2 was paralleled by respective negative controls (IDs: J, L).

### 2.3. Sample Preparation

All materials, which were in contact with the tissue samples taken, such as glass flasks (DURAN^®^, Schott, Germany) or metal spatulas, were rinsed multiple times with ultra-pure water (Millipore SAS, 67120 Molsheim, France), covered with aluminum foil, and dried in a sterile oven at 100 °C for 2 h prior to use. Nitrile gloves (VWR International GmbH, Darmstadt, Germany) were worn during all preparations. Aqua Kem Blue solution (Thetford B.V., AP Etten-Leur, The Netherlands) was filtered using a 34 µm stainless steel filter and diluted with ultra-pure water according to the manufacturer’s recommendations before it was used for storage and analysis of tissue samples.

Whole placental biopsies were cut with a metal knife and scissors into 7–8 pieces of 1 × 1 × 1 cm in size outside the operating area under a negatively pressurized sterile bench and were finally transferred into the prepared cleaned glass bottles. For the placental 7–8 core material samples (ID: I), the outer layers of tissue were removed with a metal knife, and only the core placental tissues were sampled.

### 2.4. Experimental Protocol for Microplastic Detection in Placentaand Meconium

After arriving at the Environment Agency Austria laboratory, the outside of the closed sample bottles was rinsed with ultra-pure water several times prior to any further manipulations to remove potential contaminations due to shipping (fragments of packaging materials, tapes, etc.). Furthermore, it was ensured that all sample preparation steps, including vacuum filtration, were performed in a cleanroom.

To maintain high analytical standards, all laboratory ware, such as beakers, flasks, test tubes, sieves, spatulas, etc., which came into contact with the samples, were rinsed with ultrapure Milli-Q water at least 3 times prior to use. All chemicals used during analysis were filtered via sieving through a 34 µm stainless steel sieve (Bückmann GmbH & Co. KG, Moenchengladbach, Germany), or through a 0.2 µm inorganic membrane (Whatman Anodisc, Sigma-Aldrich Chemie Gmbh, Munich, Germany), and all laboratory utensils were kept covered to prevent contamination by airborne microplastics.

The samples underwent a two-step process to separate plastic and non-plastic compounds. First, to isolate particles greater than 50 µm, each sample was sieved through a 50 µm stainless steel sieve (Bückmann GmbH & Co. KG, Germany) and rinsed out into a beaker with hydrogen peroxide (H_2_O_2_ 30%, 25 °C). It takes time (about 5 weeks for meconium and up to 7 weeks for placenta samples) and requires additional dosage or refreshing of the solution to completely eliminate the organic matter from the samples without damaging the microplastics. Second, the residuals were sieved again using a 50 µm filter (Bückmann GmbH & Co. KG, Germany) and then carefully placed into sodium hydroxide (NaOH 0.05 M, 25 °C) solution. NaOH is ideally suited to gently saponify fatty compounds from samples of this kind. Finally, the samples were quantitatively transferred onto a 50 µm stainless steel sieve, thoroughly rinsed, and flushed out with ultrapure water into a pre-cleaned beaker. Obtained solution was vacuum filtered through a 47 mm Anodisc membrane filter (Whatman, pore size 0.2 µm), which retains the particulate. Membrane filters, disposable Pasteur pipettes, glass filtering funnels, and Petri dishes were rinsed with ultrapure water each time before use. Once the particles were reliably filtered on the membrane and the vacuum was released, the filters containing them were placed in lidded Petri dishes and dried at 60 °C overnight before FTIR analysis.

The FTIR measurements were performed on a Perkin Elmer Frontier MIR/FIR Spectrometer coupled with a Spotlight 400 FT-IR Imaging System (Perkin Elmer, Traiskirchen, Austria). Infrared spectra were recorded in transmission mode in the wavenumber range of 4000 cm^−1^–1250 cm^−1^ at a spectral resolution of 16 cm^−1^, applying 2 co-added scans (pixel size 25 µm). The obtained data were further analyzed by a SpectrumIMAGE™ software (version R1.10, Copyright 2019 PerkinElmer, Inc., Traiskirchen, Austria), providing detailed information on the identity, quantity, and size of microplastic particles. The spectra of particles of interest were compared with an in-house library and assigned by expert knowledge to reference spectra of the plastic polymers. Their number was limited to 10 common plastics: polyethylene (PE), polypropylene (PP), polyvinyl chloride (PVC), polystyrene (PS), polyethylene terephthalate (PET), polyamide (PA), polyurethane (PU), polycarbonate (PC), polymethyl methacrylate (PMMA), and polyoxymethylene (POM).

In addition to screening for MPs, particle numbers were also determined in maternal stool (ID: F), as previously described [7]. In order to ensure that the used chemicals do not negatively affect appearance, weight, or IR spectrum of microplastics, 0.2–1 mm plastic fragments of PE, PP, PVC, PS, PET, PA, PU, PC, PMMA, and POM were tested before and after the sample pre-treatment process. Only low levels of modification or damage but no fragmentation of particles was observed. One additional procedural blank sample (running through the analytical method) was processed and analyzed together with the samples to cover all potential contamination sources.

### 2.5. Quality Assurance and Quality Control

All stages of sample preparation and manipulations were performed at the Environment Agency Austria exclusively under cleanroom conditions. The samples were handled in an isolated, clean, and windowless room with separate ventilation (incoming air is additionally filtered through a 34 µm stainless steel mesh) and restricted access, which was dedicated for MP analysis. Exclusive use of non-plastic materials, wearing cotton lab coats, proper cleaning of equipment, surface, etc., were ensured by specially trained staff. Cleanroom sticky mats were used to pull dirt, debris, and other impurities off shoes and help prevent these contaminants from entering inside. The MP cleanroom is additionally equipped with a laminar flow bench (EHRET Reinraumtechnik, Typ ET 130 H, EHRET GmbH, Mahlberg, Germany) to enable a more efficient sample preparation. A vacuum filtration apparatus made of glass (Whatman, GV 025/0, Ref. No.10441000) is arranged inside the laminar flow bench and connected to an electric pump which is installed outside the cleanroom to avoid the airflow coming from it. Constant internal monitoring of chemicals, sieves, and air controls is carried out in addition to analyzing the procedural blanks. Such monitoring is implemented as a complementary measure to oversee the real level and identify the potential source of contamination following the newest criteria on QA/QC in MP analytics published recently [27].

## 3. Results

All results for MPs in the placenta, meconium, stool, and contamination controls are summarized in Table 1 and Table 2. In the initial phase 1, screening for MPs in placental tissue was positive for PE, PP, and PU (IDs: A + B, Table 1A). The negative placental control (ID: C) was screened negative for MPs. Additionally, meconium samples were screened positive for PE (ID: D); however, the meconium negative control tested positive for PP (ID: E).

In the maternal stool sample, approximately 2 PE and 1 PS microplastic particles were found per 20 g of stool (ID: F, Table 1B). No other MP materials were detectable in the maternal stool. However, the negative control for maternal stool screened positive for the presence of PE, PP, and PS (ID: G, Table 1A).

In addition, 16 probes from plastic-containing materials from the surgical equipment and lab were screened for MPs as contamination controls (ID: 1–16, Table 1C). In these controls, PE, PP, and PET were commonly detected. PU, however, was not detected in any of the contamination control samples.

In phase 2 of the study, no MPs were found after intense washing of the placenta (ID: H, Table 2); however, PP was detected in the core sample (ID: I, Table 2). Yet, the negative control tested positive for PS (ID: K, Table 2). The meconium samples in phase 2 were screened positive for PP and PS (ID: K, Table 2, Figure 3 and Figure 4), while the negative control revealed no MPs (ID: L, Table 2). The investigation for airborne fallout of MPs in the operating theater showed the presence of PU in the air (ID: M, Table 2).

## 4. Discussion

To the best of our knowledge, this is the first pilot study to evaluate the possibility of screening for MPs > 50 µm in near-term human placenta and meconium in a “real live clinical setting during cesarean delivery”. MPs were detected in the placenta and meconium. However, critical evaluation of the extensive contamination controls revealed possibly sample contamination, and despite all precautions, screening for MPs in clinical settings remains challenging, and study results, in general, should be interpreted with caution.

The involuntary exposure of MPs via food and respiration seems to be certain [4]. The intrauterine transmission via the placenta to the fetus is likely, and potential risks during fetal development need to be acknowledged [22]. MPs accumulation in the placenta and even transplacental transport has been demonstrated, but so far only in ex vivo models [6,21,22]. Yet, these studies bear some limitations. Results need to be scrutinized whether the evidence of MPs in organ tissues has been demonstrated in vivo or ex vivo, if projected rather than measured values are presented, and if the effects are observed in a MP challenge setting or under “physiologic” circumstances. The question of whether MP actually accumulates in the placenta during pregnancy is of the highest relevance since microparticles, in general, may alter several cellular regulating pathways in the placenta and therefore may influence placental and fetal development [28]. There is only one study to our knowledge which screened for MPs in the placenta, but studied vaginal delivery [25]. While the method of delivery indeed has an impact on the child, including its microbiome [29], it has not yet been established if this holds true for potential MP contamination during placenta sampling.

In our study, we detected MPs in the placenta and stool in a real-life setting. Therefore, we had to develop a new methodology. The protocols focused on avoiding potential contamination with MPs, and thus multiple potential contaminants were examined for their plastic content and compared to the findings in the placenta, meconium, and maternal stool. Cesarean delivery and breech delivery allowed to better control for possible plastic contamination. Biological samples were collected in prewashed metal containers and immediately prepared for transport. In addition, negative samples were screened for MPs to rule out contamination during the tissue-preparation process. Maternal stool was collected according to previously established protocols [7].

Ragusa et al. recently described 12 microplastic fragments, ranging from 5 to 10 μm in size, in 4 out of 6 placentas delivered vaginally, and identified polypropylene most commonly [25]. Such small foreign particles were previously detected in human organs [30]; however, the way they are internalized is still unclear. In the present study, placenta, meconium, and maternal stool were collected and screened for MPs > 50 µm content. In addition to polypropylene, polyethylene and polyurethane were found in whole-tissue blocks of placenta samples. It is important to note that the placenta control sample tested negative for MPs, ruling out contamination during the tissue-preparation process. Meconium tested positive for polyethylene; unfortunately, the meconium control samples also tested positive for polypropylene, indicating possible contamination here. The 16 contamination controls all screened positive for several types of polymers, including polyethylene, polypropylene, polyvinylchloride, polystyrene, polyethylene terephthalate, polyamide, polyurethane, polycarbonate, polymethylmethacrylate, polyoxymethylene, ethylene-propylene-diene rubber copolymer, and acrylonitrile-butadiene rubber. Interestingly, the detected polyurethane in the placenta tissue was not found in any of the contamination control samples and may indicate actual accumulation in the placenta. Maternal stool samples were positive for polyethylene and polystyrene as described previously [7].

To further investigate the possible origin of contamination, the airborne fallout was investigated in phase 2 of the study. The collection of an air probe for 10 min during cesarean delivery showed a positive test for polyurethane, indicating that the detection of PU in the placenta might be due to contamination from airborne fallout in phase 1. To avoid placental contamination in phase 2, the placental tissue block was either washed several times with MilliQ water before it was stored in the transport bottles, or the outer surface of the placenta was carefully removed with a metal scalpel and only the placenta core was sent for MPs screening. Here, the placenta sample tested negative for MPs after washing, but the core sample tested positive for polypropylene. The smaller the particles that are detected, the more difficult it is for contamination to be controlled [31]. Moreover, the breakdown of larger MP particles may lead to artificially high MP counts [32,33]. In view of the particle size of >50 µm examined by us and despite numerous methodological measures to avoid contamination, the previously described detection of microplastic particles with a size of 5–10 µm [25] should be interpreted with caution.

Study limitations are obvious due to the high risk of environmental contamination with MPs in a clinical setting. We have recently published a study on the non-existence of the fetal gut microbiome [29]. The study design was very similar to the present study, and the data suggest that microbiological colonization occurs during birth via maternal skin/vaginal/fecal seeding or post-birth via environmental seeding. Even during the cesarean section, it was very difficult to avoid microbiological contamination. Although data exists, the likelihood of contamination with MPs during vaginal delivery must be considered as a greater source of contamination than the collection of samples during cesarean section.

Analysis of microplastics by Raman microspectroscopy offers the advantage that significantly smaller particles can be examined, but the areas that can be examined and thus the amount of sample that can be used are very small. This raises the question of the representativeness of the examined sample. Fourier-transform infrared microspectroscopy in transmittance mode is hampered by methodological complexities and a lack of standardized approaches for sampling, sample preparation, identification, and quantification of MPs in samples of this kind. The method requires infrared-transparent filter substrates made of non-polymer materials (e.g., Anodic alumina matrix membrane), which in turn restricts the wavenumber range available for spectral acquisition to 4000–1250 cm^−1^. FTIR determines the size, number, shape, and chemical composition of the particles and provides information about their distribution on the filter surface. However, it is not possible to estimate the mass of MPs within the sample using this method. The entire process of identification and quantification of MPs is time- and cost-consuming and therefore limited to the 10 polymer types.

## 5. Conclusions

First reports now indicate a potential intrauterine exposure with MP, yet readouts are prone to contamination. We now established for the first time a thorough protocol for the detection of MPs in the human placenta and fetal meconium in a real-life clinical setting. In a pilot study, we found MPs > 50 µm in placenta and meconium acquired from cesarean breech deliveries. However, critical evaluation of potential contamination sources is pivotal and may guide future clinical studies to improve correct detection of MPs in organ tissue.

## Figures and Tables

**Figure 1 pharmaceutics-13-00921-f001:**
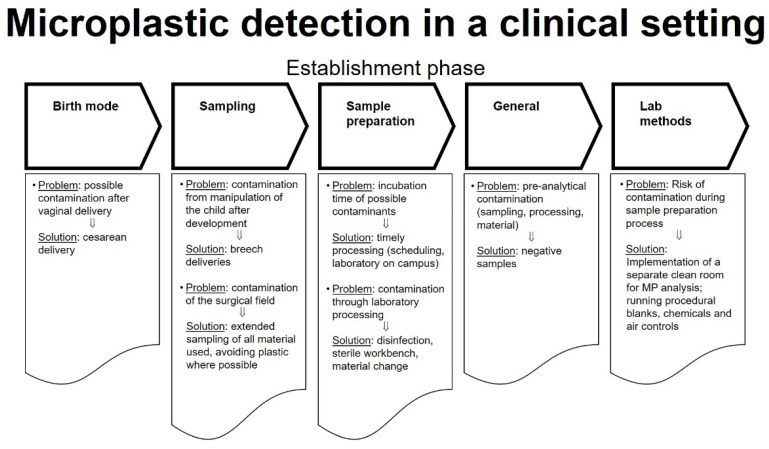
Microplastic detection in a clinical setting, “Establishment phase”.

**Figure 2 pharmaceutics-13-00921-f002:**
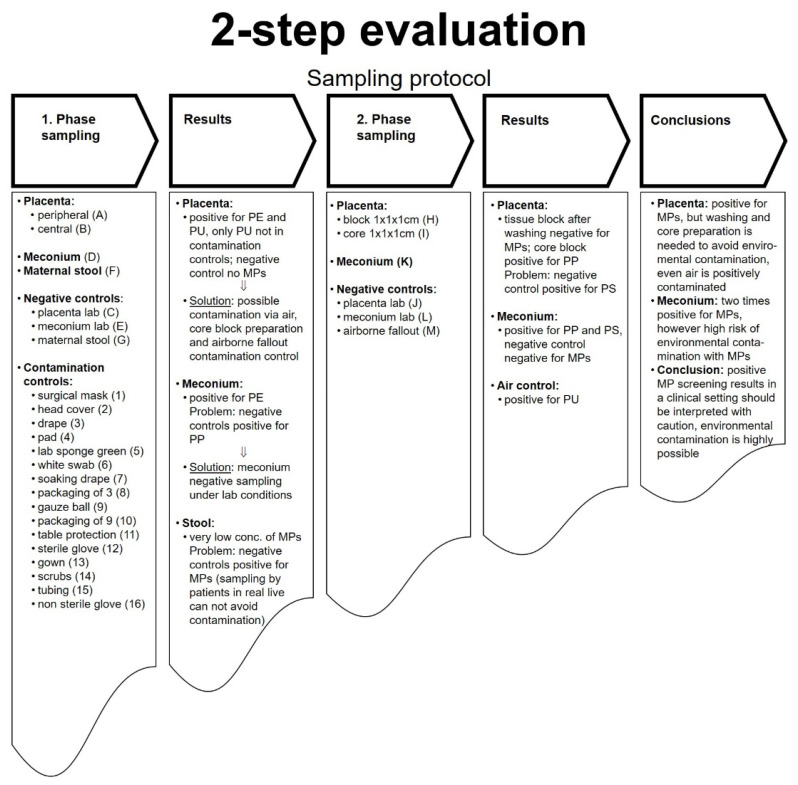
2-step evaluation protocol. In phase 1, placental tissues, meconium, and maternal stool, negative controls, and contaminations controls were collected and analyzed for MPs. Negative controls include testing for MPs > 50 µm of the instruments used during the preparation of the tissues (placenta, meconium, maternal stool) without the actual tissues. To rule out possible contamination with plastic during the sampling and preparation steps, in phase 2, the placenta blocks were either intensively washed or only the “core” placental tissue was collected. Next to meconium, airborne fallout was sampled.

**Figure 3 pharmaceutics-13-00921-f003:**
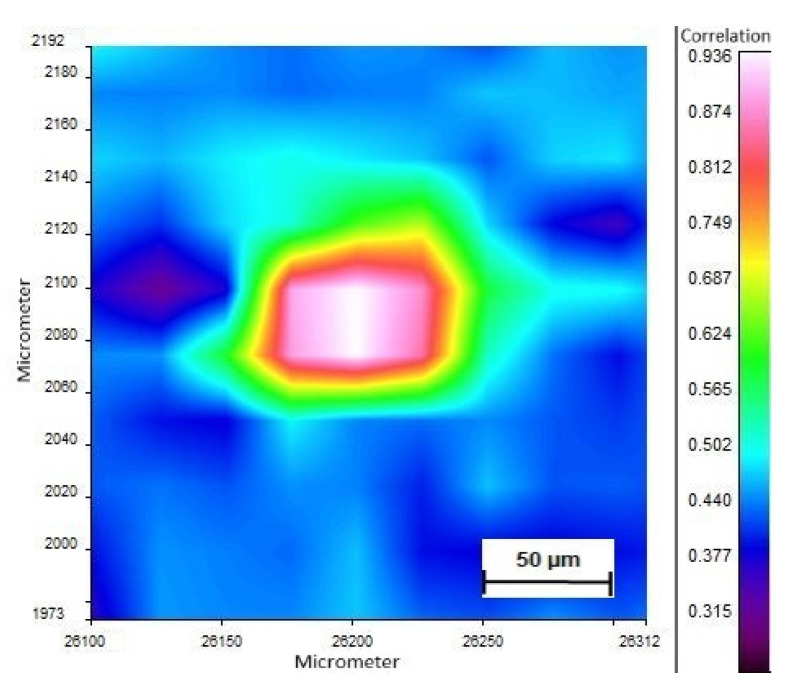
Polystyrene particle detected in the meconium sample. FTIR image displays the correlation value with polystyrene.

**Figure 4 pharmaceutics-13-00921-f004:**
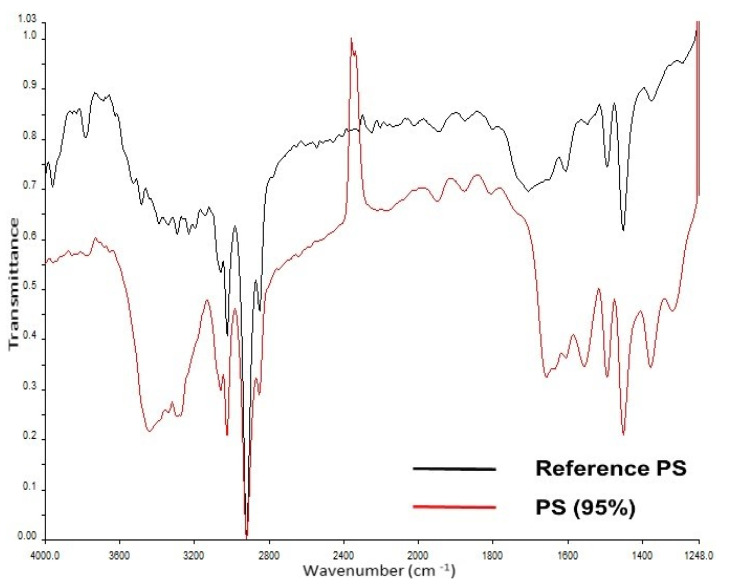
Measured IR spectrum in the sample (in red) and reference spectrum from in-house library (in black) for polystyrene.

**Table 1 pharmaceutics-13-00921-t001:** (**A**–**C**): First phase screening for microplastic particles in (**A**) human placenta, (**A**) meconium, (**B**) maternal stool, and (**C**) contamination controls (0.05–0.5 mm). Negative controls include screening for MPs of sample preparation instruments and storage containers without the actual tissues (placenta, meconium, maternal stool). PE = polyethylene, PP = polypropylene, PVC = polyvinylchloride, PS = polystyrene, PET = polyethylene terephthalate, PA = polyamide, PU = polyurethane, PC = polycarbonate, PMMA = polymethylmethacrylate, POM = polyoxymethylene, EPDM = ethylene-propylene-diene rubber copolymer, NBR = acrylonitrile-butadiene rubber; * limit of quantification (LOQ) = 0.48.

**(A)**				
**ID**	**Sample**	**Description**	**Screening for MPs**	**Type**
A	Placenta	sampled from the peripheral regions	positive	PE, PP
B	Placenta	sampled from central regions	positive	PE, PU
C	Placenta	negative control, sampled in lab	negative	-
D	Meconium	sampled on metal	positive	PE
E	Meconium	negative control, sampled in lab	positive	PP
G	Stool	negative control for stool post partum	positive	PE, PP, PS
**(B)**				
**ID**	**Sample**	**Description**	**Screening for MPs**	**No/10 g ***
F	Stool	maternal stool post partum	PE	0.96
			PP	<0.48
			PVC	<0.48
			PS	0.48
			PET	<0.48
			PA	<0.48
			PU	<0.48
			PC	<0.48
			PMMA	<0.48
			POM	<0.48
**(C)**				
**ID**	**Sample**	**Description**	**Screening for MPs**	**No/10 g**
1	surgical mask	surgical mask, white, Mölnycke Health Care AB, Göteborg, Sweden	fleece plastic around wire	PP PE
2	head cover	surgical head cover, green, FarStar medical GmbH, Barsbüttel, Germany	green material white material yarn	PP PET PET
3	drape	cesarean section drape with sponge, Medline International France, Chateaubriant, France	film sponge	PE copopolymere
4	pad	3-layer cesarean pad, Medline Industries Inc., Northfield, USA	fleece green foil white filling	PP PE cellulose
5	lab sponge green	abdominal swab, green, Allmed Medical Products Co., Zhijiang City, China	negative	cellulose
6	white swab	two-layer white paper swabs from surgical operation set, Allmed Medical Products Co., Zhijiang City, China	negative	cellulose
7	soaking drape	Two-layer additional side drapes, 3M Deutschland GmbH, Neuss, Germany	Film fleece	PE PP
8	packaging	packaging of cover sheet (3), Allmed Medical Products Co., Zhijiang City, China	rough side smooth side	cellulose polysiloxane
9	gauze ball	white surgical gauze ball, Nobamed Paul Danz AG, Wettler, Germany	negative	cellulose
10	packaging	packaging of cover sheet (9), Allmed Medical Products Co., Zhijiang City, China	film	PE
11	table protection	surgical table covering, blue, Medline International France, Chateaubriant, France	film	PE
12	sterile glove	sterile surgical glove, CardinalHealth, Waukegan, USA	rubber	EPDM
13	gown	dark blue operating gown, Medline International France, Chateaubriant, France	fleece	PP
14	scrubs	blue scrub, SITEX GmbH, Minden, Germany	solid fabric	PET
15	tubing	operating suction tubing, Pennine Healthcare, Derby, UK	pipe	PET
16	non-sterile glove	non-sterile lab glove, Medical Device Safety Service GmbH, Hannover, Germany	rubber	NBR

**Table 2 pharmaceutics-13-00921-t002:** Second phase screening for microplastic particles in human placenta, meconium and airborne fallout (0.05–0.5 mm). Negative controls include screening for MPs of the sample preparation instruments and storage containers without the actual tissues (placenta, meconium, maternal stool). PP = polypropylene, PS = polystyrene, PU = polyurethane.

ID	Sample	Description	Screening for MPs	Type
H	Placenta	block 1 × 1 × 1 cm	negative	-
I	Placenta	core block 1 × 1 × 1 cm	positive	PP
J	Placenta	negative control, sampled in lab	positive	PS
K	Meconium	sampled on metal	positive	PP, PS
L	Meconium	negative control, sampled in lab	negative	-
M	airborne fallout	5 min air probe from the operating theatre	positive	PU

## Data Availability

Not applicable.

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
