# Peer review of "Detection of Microplastic in Human Placenta and Meconium in a Clinical Setting"

_pharmaceutics, 2021, doi:10.3390/pharmaceutics13070921_

Round 1

Reviewer 1 Report

In this study, a FTIR method was used to detect the presence of microplastics human placenta and meconium. The significance of this study is positive. However, some issues need to be improved before publication. Major revision is suggested. 1. Authors should follow the guidelines for writing papers, which can be found in published studies. 2. The authors are failed on highlight the value of this study in current form. 3. Considering the location of the authors' experiments, I can believe the authors' QA/QC are qualified. However, this part is important enough for the authors to describe it in separate subsections. 4. Table 1, rough forms need to be improved. 5. Figure 2, bad color matching makes the words hard to read. 6. Figures of microplastics should be provided: before and after extraction (If ethical requirements are allowed).

Author Response

Reviewer #1:

In this study, a FTIR method was used to detect the presence of microplastics human placenta and meconium. The significance of this study is positive. However, some issues need to be improved before publication. Major revision is suggested.

  1. Authors should follow the guidelines for writing papers, which can be found in published studies. Comment: We have structured the article according to the Reporting Guidelines Checklist from the following article: Cowger W, Booth AM, Hamilton BM, et al. Reporting Guidelines to Increase the Reproducibility and Comparability of Research on Microplastics. Applied Spectroscopy. 2020;74(9):1066-1077.

  1. The authors are failed on highlight the value of this study in current form.

Comment: We have modified the conclusions for clarification (lines 361-367).

  1. Considering the location of the authors' experiments, I can believe the authors' QA/QC are qualified. However, this part is important enough for the authors to describe it in separate subsections. Comment: We have added an extra paragraph on “Quality assurance and quality control” under chapter 2.5 into the manuscript (lines 200-215).

  1. Table 1, rough forms need to be improved.

Comment: Table layout was modified according to the journals style.

  1. Figure 2, bad color matching makes the words hard to read.

Comment: Figure 1 and 2 are now in black and white for better visibility.

  1. Figures of microplastics should be provided: before and after extraction (If ethical requirements are allowed).

Comment: Thank you very much for this important comment. We have added Figure 3 and 4 to the manuscript.

Reviewer 2 Report

Dear Authors,

I reported my comments on the manuscript titled “Detection of microplastic in human placenta and meconium in a clinical setting”.

I think that the manuscript is interesting. References list appears up to date and appropriate.

Before the acceptance I suggest some major revisions.

At line 254-257, you say that with vaginal delivery is impossible to completely rule out contamination of external microplastics, and affirmed that the delivery through caesarean section reduce the possibility of contamination. I am not in accordance with this fact and I think that is only an inference, with no scientific basis. In this doubtful consideration you put in references the recent article of Ragusa et al. 2021. I am an authors of this cited article, and I can, with no doubts, say that the modality of delivery is irrelevant toward the reduction of contamination, but the only important thing, is follow a rigorous plastic-free protocol for the sampling. The fact that you recruited only two patients from a caesarean section is not a merit, is a fact, so I suggest to remove this consideration, and leave only the sentence: "There is only one study to our knowledge, which screened for the presence of microplastcs in placenta".

After this consideration I suggest modify the figure 1, because the mode of birth is not a problem with a solution to find. Moreover in the figure 1 a second aspect is the sampling. For this reason the authors decided to recruit two women with scheduled caesarean section for breech presentation. For you the breech presentation reduce the possibility to contamination from manipulation. This is not true, there are not literature data or evidences on that. Second aspect is "the contamination of the surgical field and the extended sampling of all material used". As previous say, the only important thing, is follow a rigorous plastic-free protocol for the sampling. So I suggest to erase this two first aspects (birth mode and sampling) from the figure 1. After that I suggest to erase the sentence at line 262-263 “Cesarean delivery and breech delivery allowed to avoid possible plastic contamination, which could occur during a vaginal birth.”, because it is not true.

What is the placenta control sample? (line 274-275). This is not easy to understand. In general the methods are not clear. This fact is a great problem because if I would want repeat the study I will not able to do this. Probably there is a nomenclature problem, I suggest to improve the readability of the methods.

You speak about the research of microplastics in the stool, but this aspect is not mentioned in the manuscript title. Why this choice? I suggest check this aspect.

At line 296-299, start a sentences of a scientific paper with “In our opinion” is not a good idea. Your personal opinion is not useful from a scientific point of view, we need facts and evidences. Moreover, all studies on this field should be interpreted with caution, not only Ragusa et al.’s study, because the possibility of contamination is always possible in spite of all efforts to avoid that. For this reason I suggest to change the sentence put in light this fact..

References list appears up to date. I suggest to check all references and use Vancouver style for them. I noted that the references are expressed in number in round brackets at the beginning of the manuscript, but with first author in round brackets in the discussion . I suggest adjust the style of references following the journal authors guidelines.

After these major revisions, I suggest to accept this manuscript.

Author Response

Reviewer #2:

1.) At line 254-257, you say that with vaginal delivery is impossible to completely rule out contamination of external microplastics, and affirmed that the delivery through caesarean section reduce the possibility of contamination. I am not in accordance with this fact and I think that is only an inference, with no scientific basis. In this doubtful consideration you put in references the recent article of Ragusa et al. 2021. I am an author of this cited article, and I can, with no doubts, say that the modality of delivery is irrelevant toward the reduction of contamination, but the only important thing, is follow a rigorous plastic-free protocol for the sampling. The fact that you recruited only two patients from a caesarean section is not a merit, is a fact, so I suggest to remove this consideration, and leave only the sentence: "There is only one study to our knowledge, which screened for the presence of microplastics in placenta".

Comment: It is well known, that the route of delivery plays a very important role for many reasons. It affects neonatal short- and long-term outcomes and plays a major role for example in the development of the neonatal and infant microbiome. Very recently, we have published a study in Nature Microbiology on the non-existence of the fetal gut microbiome (1). The study design was very similar to the present study and one major point, beside the implementation of several negative controls, was the collection of meconium of fetuses in breech presentation during c-section, in order to avoid possible environmental bacterial contamination, such as during vaginal delivery. Our data suggests, that colonization occurs during birth via maternal skin/vaginal/fecal seeding or post-birth via environmental seeding. However, as discussed, even during c-section it was very difficult to avoid microbiological contamination, in at least half of the cases. Transferring those results to our study on MP detection, the likelihood of contamination with MP during vaginal delivery must be considered as a source of contamination and that’s why we choose the cesarean section as alternative route of delivery. Since MP can be detected on all kinds of surfaces, it is very likely that also the vaginal epithelium most likely contains MP. Next to underwear, panty liners, intercourse or toilet paper, transvaginal ultrasound scans during pregnancy and several digital examinations during delivery are possible sources of MP contamination. Therefore, a vaginally delivered placenta most likely gets into contact with those MP contaminants, even when undertaking best attempts to apply a “plastic-free” protocol. We agree with the reviewer that contamination cannot completely be ruled out, even by cesarean delivery and therefore amended the manuscript and additionally addressed this important point critically in our study limitations (lines 280, 293, 298, 336-343).

2.) After this consideration I suggest modify the figure 1, because the mode of birth is not a problem with a solution to find. Moreover in the figure 1 a second aspect is the sampling. For this reason the authors decided to recruit two women with scheduled caesarean section for breech presentation. For you the breech presentation reduce the possibility to contamination from manipulation. This is not true, there are not literature data or evidences on that.

Comment: Please see above.

3.) Second aspect is "the contamination of the surgical field and the extended sampling of all material used". As previous say, the only important thing, is follow a rigorous plastic-free protocol for the sampling. So I suggest to erase this two first aspects (birth mode and sampling) from the figure 1. After that I suggest to erase the sentence at line 262-263 “Cesarean delivery and breech delivery allowed to avoid possible plastic contamination, which could occur during a vaginal birth.”, because it is not true.

Comment: Please see above.

4.) What is the placenta control sample? (line 274-275). This is not easy to understand. In general the methods are not clear. This fact is a great problem because if I would want repeat the study I will not able to do this. Probably there is a nomenclature problem, I suggest to improve the readability of the methods.

Comment: We have improved the readability of the respective methods. Negative controls (IDs: C=placenta, E=meconium, G=stool; table 2) included the preparation instruments and storage containers without actual tissues and were collected in the lab.

 5.) You speak about the research of microplastics in the stool, but this aspect is not mentioned in the manuscript title. Why this choice? I suggest check this aspect.

Comment: Our study focuses on the detection of MP in human placenta and meconium, which also represents the main novelty of our research. Detecting the presence of MP in human stool has been evaluated previously (2), and was used in our study primarily as a positive control. Hence, we prefer to not change the title of our study.

6.) At line 296-299, start a sentences of a scientific paper with “In our opinion” is not a good idea. Your personal opinion is not useful from a scientific point of view, we need facts and evidences. Moreover, all studies on this field should be interpreted with caution, not only Ragusa et al.’s study, because the possibility of contamination is always possible in spite of all efforts to avoid that. For this reason I suggest to change the sentence put in light this fact..

Comment: Thank you for pointing out this detail. We have modified the sentence. Yet, a crucial part of any scientific discussion includes a critical analysis of own and historical data. We are sure, the reader will benefit from this and hope that future studies will be able to further elucidate this fascinating topic.

7.) References list appears up to date. I suggest to check all references and use Vancouver style for them. I noted that the references are expressed in number in round brackets at the beginning of the manuscript, but with first author in round brackets in the discussion. I suggest adjust the style of references following the journal authors guidelines.

Comment: Thank you, this has been modified.

8.) After these major revisions, I suggest to accept this manuscript.

Comment: We are grateful for this comment and hope that we could clarify all points mentioned above.

References:

  1. Kennedy KM, Gerlach MJ, Adam T, Heimesaat MM, Rossi L, Surette MG, et al. Fetal meconium does not have a detectable microbiota before birth. Nat Microbiol. 2021.
  2. Schwabl P, Köppel S, Königshofer P, Bucsics T, Trauner M, Reiberger T, Liebmann B. Detection of Various Microplastics in Human Stool: A Prospective Case Series. Ann Intern Med. 2019;171(7):453-457.

Reviewer 3 Report

First, congratulations for such a nice manuscript. It is very interesting the fact you describe in the article. Howerver, I would like to ask you to describe better the way you put the filtrate containing the microparticles into a membrane for FTIR analysis. Please, modify the manuscript accordingly in order to improve the quality. 

Author Response

Reviewer #3:

1.) First, congratulations for such a nice manuscript. It is very interesting the fact you describe in the article. However, I would like to ask you to describe better the way you put the filtrate containing the microparticles into a membrane for FTIR analysis. Please, modify the manuscript accordingly in order to improve the quality. 

Comment: A detailed description has been added to the manuscript, please see lines 160-167.

Round 2

Reviewer 1 Report

I think the current version of manuscript meets the publishing standards already .

Author Response

Thank you very much! A naitive speaker (American) has now revised the manuscript and changes are indicated in yellow.

Reviewer 2 Report

Dear Authors,

I have read your answers, and I have some comments.

I know very well the fact that “the route of delivery plays a very important role for many reasons. It affects neonatal short- and long-term outcomes and plays a major role for example in the development of the neonatal and infant microbiome.” But the true is that there is no evidence that the route of delivery can influences the risk of microplastics contamination. You know that fact, indeed, you say: “Since MP can be detected on all kinds of surfaces, it is very likely that also the vaginal epithelium most likely contains MP”. “likely” mean that it is probably but not sure. So, I formally ask to remove from the article all sentences about the fact that take samples from a cesarean section is better, compare to take sample from a vaginal delivery. This means, remove the sentence in line 294 and change the sentences at line 299-300 “Cesarean delivery and breech delivery allowed to reduce possible plastic contamination, which could occur during a vaginal birth”. This consideration is true for other aspects, well described by you in the responds to my comments but is not yet demonstrated for microplastics contamination. So, you have the duty to present your results without discredit the results of others, unless there is scientific evidence to the contrary and in this case, there is no evidence that vaginally collected placentas are more or less contaminated than placentas collected after a cesarean

At line 331-332 “The smaller the particles that are detected, the more difficult it is for contamination to be controlled” is necessary add some literature references that sustain this sentence. If there are not references about this, I suggest changing the sentence considering the same duty asked before to present your results without discredit the results of others.

Author Response

Please find attached our comments:

1.)  So, I formally ask to remove from the article all sentences about the fact that take samples from a cesarean section is better, compare to take sample from a vaginal delivery. This means, remove the sentence in line 294 and change the sentences at line 299-300 “Cesarean delivery and breech delivery allowed to reduce possible plastic contamination, which could occur during a vaginal birth”. This consideration is true for other aspects, well described by you in the responds to my comments but is not yet demonstrated for microplastics contamination. So, you have the duty to present your results without discredit the results of others, unless there is scientific evidence to the contrary and in this case, there is no evidence that vaginally collected placentas are more or less contaminated than placentas collected after a cesarean.

Comment: We have modified line 294: "

There is only one study to our knowledge, which screened for MPs placenta, but which were delivered vaginally (25).  While the method of delivery indeed impacts on the child, including its microbiome (Reference: T. Braun, Nature Microbiology, 2021), it has not yet been established if this holds true for potential MP contamination during placenta sampling."

We have modified line 299: "

Cesarean delivery and breech delivery allowed to better control for possible plastic contamination."

2.) We have added references to line 331.